# Autoimmunity and Autoinflammation: Relapsing Polychondritis and VEXAS Syndrome Challenge

**DOI:** 10.3390/ijms25042261

**Published:** 2024-02-13

**Authors:** Anca Cardoneanu, Ioana Irina Rezus, Alexandra Maria Burlui, Patricia Richter, Ioana Bratoiu, Ioana Ruxandra Mihai, Luana Andreea Macovei, Elena Rezus

**Affiliations:** 1Discipline of Rheumatology, Medical Department II, University of Medicine and Pharmacy “Grigore T Popa”, 700115 Iasi, Romania; anca.cardoneanu@umfiasi.ro (A.C.); maria-alexandra.burlui@umfiasi.ro (A.M.B.); patricia.richter@umfiasi.ro (P.R.); ioana.bratoiu@umfiasi.ro (I.B.); ioana-ruxandra_mihai@umfiasi.ro (I.R.M.); luana.macovei@umfiasi.ro (L.A.M.); 2Clinical Rehabilitation Hospital, 700661 Iasi, Romania; 3Discipline of Radiology, Surgery Department II, University of Medicine and Pharmacy “Grigore T Popa”, 700115 Iasi, Romania; rezus_ioana-irina@d.umfiasi.ro

**Keywords:** relapsing polychondritis, inflammation, cartilage, anti-collagen antibodies, VEXAS syndrome

## Abstract

Relapsing polychondritis is a chronic autoimmune inflammatory condition characterized by recurrent episodes of inflammation at the level of cartilaginous structures and tissues rich in proteoglycans. The pathogenesis of the disease is complex and still incompletely elucidated. The data support the important role of a particular genetic predisposition, with HLA-DR4 being considered an allele that confers a major risk of disease occurrence. Environmental factors, mechanical, chemical or infectious, act as triggers in the development of clinical manifestations, causing the degradation of proteins and the release of cryptic cartilage antigens. Both humoral and cellular immunity play essential roles in the occurrence and perpetuation of autoimmunity and inflammation. Autoantibodies anti-type II, IX and XI collagens, anti-matrilin-1 and anti-COMPs (cartilage oligomeric matrix proteins) have been highlighted in increased titers, being correlated with disease activity and considered prognostic factors. Innate immunity cells, neutrophils, monocytes, macrophages, natural killer lymphocytes and eosinophils have been found in the perichondrium and cartilage, together with activated antigen-presenting cells, C3 deposits and immunoglobulins. Also, T cells play a decisive role in the pathogenesis of the disease, with relapsing polychondritis being considered a TH1-mediated condition. Thus, increased secretions of interferon γ, interleukin (IL)-12 and IL-2 have been highlighted. The “inflammatory storm” formed by a complex network of pro-inflammatory cytokines and chemokines actively modulates the recruitment and infiltration of various cells, with cartilage being a source of antigens. Along with RP, VEXAS syndrome, another systemic autoimmune disease with genetic determinism, has an etiopathogenesis that is still incompletely known, and it involves the activation of the innate immune system through different pathways and the appearance of the cytokine storm. The clinical manifestations of VEXAS syndrome include an inflammatory phenotype often similar to that of RP, which raises diagnostic problems. The management of RP and VEXAS syndrome includes common immunosuppressive therapies whose main goal is to control systemic inflammatory manifestations. The objective of this paper is to detail the main etiopathogenetic mechanisms of a rare disease, summarizing the latest data and presenting the distinct features of these mechanisms.

## 1. Introduction

Relapsing polychondritis (RP) is a systemic immune-mediated inflammatory disease that presents numerous recurrent inflammatory episodes at the level of cartilaginous structures and tissues rich in proteoglycans [1,2]. Thus, the main clinical manifestations presented by patients include inflammation of the ear, nose and tracheobronchial tree, as well as damage to other structures, such as the joints, skin, eye and cardiovascular or renal system [3,4]. The first attestation of the disease was made in 1923 when it was described as “polychondropathia” [5]. Then, in 1960, following an analysis of 12 cases, Pearson named the disease “relapsing polychondritis”, the name used today [6]. RP has an insidious onset, having a chronic evolution with periods of remission and exacerbation [7]. The pathogenesis of the disease is not yet fully known, but the published data place it in the category of immune-mediated diseases [8]. Approximately 30% of RP cases are associated with other autoimmune diseases, being diagnosed as overlap syndromes [9]. Associations between RP and rheumatoid arthritis, spondylarthritis and collagen diseases (such as systemic lupus erythematosus, dermatomyositis, systemic scleroderma or systemic vasculitis) have been described [8,10,11,12]. Special attention has been paid to the association between RP and the presence of malignancies, especially myelodysplastic syndrome, which can be present in up to 27% of cases [13,14].

## 2. Pathogenetic Mechanisms in RP

Although the etiopathogenic mechanisms of RP are still incompletely understood, the data that we have so far support the important role of a predisposing genetic background on which trigger factors act, thus initiating autoimmune phenomena [15]. There are no data to support the hereditary transmission of the disease [1]. The immune mechanism involved in the development of RP is complex and involves both cellular and humoral immunity [1]. Figure 1 summarizes the main pathogenic mechanisms involved in the occurrence of inflammation and cartilage damage in RP.

### 2.1. Genetic Susceptibility in RP

Numerous studies have focused on analyzing genetic susceptibility in RP, with the role of human leukocyte antigen (HLA) being recognized in the occurrence of autoimmune diseases [15]. HLA-DR4 is an allele that confers a major risk of RP occurrence, but there is no predominance of a certain subtype of DR4 alleles in this condition [16,17]. Considering these data, it is suggested that RP can be considered a different condition from other autoimmune diseases, such as systemic lupus erythematosus or rheumatoid arthritis [18]. Table 1 shows the main genes involved in the development of RP and the clinical manifestations directly correlated with this particular genetic susceptibility.

A German study analyzed the roles of HLA-DRB1, DQ and class I alleles in a cohort of 41 patients with RP compared to 204 healthy individuals [16]. The frequency of HLA-DR4 was significantly higher in the group of RP patients; no predisposition for any class I alleles was highlighted [16]. Other data were published following an analysis that included HLA-DQ6/8 double-transgenic mice that developed auricular chondritis, supporting the role of genetic background in the pathogenesis of this disease [19]. 

Moreover, Zeuner et al. highlighted that there is a negative correlation between the presence of HLA-DR6 and the severity of organ damage in RP; HLA-DR6-positive patients also presented a higher median age at disease onset [17]. Another analysis of the important role of HLA class II DQ alleles included 64 RP cases, and they showed a higher frequency of DQB1*0601, DQA1*0103 and DQA1*0301 than the control group [20]. 

More recent data have highlighted a new genetic susceptibility in RP cases that distinguishes this disease from other chronic autoimmune rheumatic disorders. The study group included 102 RP cases and 1000 healthy people. The results reinforced the importance of HLA class II in susceptibility to the disease; in addition, HLA-DRB1*16:02, HLA-DQB1*05:02 and HLA-B*67:01, in linkage disequilibrium with each other, significantly participates in the genetic susceptibility of RP [21].

Recently, a syndrome was described that occurs only in men and that includes inflammation of the cartilage, blood vessels and skin, as well as hematological manifestations [15]. This condition is called VEXAS syndrome, with the acronyms referring to its main characteristics: vacuoles, E1 enzyme, X-linked, autoinflammatory manifestations and somatic manifestations [15,22]. It is very possible that more than half of the individuals diagnosed with RP who have associated myelodysplastic syndrome actually have VEXAS syndrome [22]. According to data from studies, patients with VEXAS-RP present somatic variants in ubiquitin-activating enzyme-1 (UBA1), and these are associated with various clinical manifestations and a higher mortality [22,23,24].

Although genetic predisposition has an important place in the pathogenesis of the disease, it seems that certain environmental factors act as triggers, favoring the development of the clinical manifestations of the disease.

### 2.2. External Triggers in RP

There are unknown external triggers that act on individuals with a genetic predisposition, causing the degradation of proteins and the release of cryptic antigens at the cartilage level, thus initiating the disease. Among these external triggers, an important role is attributed to mechanical, chemical and infectious factors. 

Each of them participates in the activation of autoimmunity in a particular way, determining the development of inflammatory clinical manifestations at the level of cartilaginous structures, such as the nose, ear, joints, eyes or respiratory tract. All these data are summarized in Table 2.

Mechanical factors include direct trauma to the cartilage; this leads to the exposure of cartilage matrix protein antigens, which are responsible for the occurrence of an autoimmune response [25]. A published case study highlighted the appearance of inflammatory changes in the nose, ears and upper respiratory tract after an ear piercing [26]. 

Chemical factors involved in the onset of the disease refer mainly to the intravenous administration of various substances that can have a toxic effect on cartilage. The data show the rapid appearance, within 24 h, of clinical manifestations similar to those of RP, such as nasal and auricular inflammation, peripheral and axial joint damage, scleritis and vestibular disorders after an intravenous injection of some toxic substances (hydrochloric acid, carburetor fluid and the waxy internal matrix of a mentholated nasal inhaler) [27]. Additionally, the rapid onset of bilateral auricular chondritis was described after the initiation of chondroitin and glucosamine therapy [28]. Moreover, an intravenous injection of papain in rabbits was associated with important clinical changes, such as ear collapse or tracheal and bronchial damage, even causing acute respiratory distress [29].

Infectious triggers can determine the activation of the immune system through molecular mimicry, which is a structural similarity between one’s own heat shock proteins (HSPs) and microbial HSPs [30]. Thus, the innate immune system is activated through the TLR (Toll-like receptor) and NLR (nucleotide-binding oligomerization domain-like receptor) signaling pathways [31]. Cases of RP associated with chronic hepatitis C virus infection have been reported in the literature [32,33]. Other data support the possible role of Mycobacterium tuberculosis in the pathogenesis of RP because an antibody linked both to HSP60 from the bacterium and to cartilage structures was highlighted [34]. Last but not least, it seems that there is a close relationship between intestinal dysbiosis and RP, as demonstrated by a metagenomic analysis. Gut dysbiosis in these patients is characterized by a numerical increase in *Ruminococcus*, *Bacteroides*, *Veillonella* and *Eubacterium* species [35]. 

### 2.3. Humoral Mediators in RP

#### Autoantibodies in RP

Anti-cartilage antibodies play an important role in the development of RP, with studies demonstrating the presence of anti-type II, IX and XI collagen antibodies in high titers [36,37]. Also, there is a direct correlation between the level of these antibodies and the activity of the disease, the antibodies being considered prognostic factors [36,38]. Numerous data support the increased level of anti-type II collagen antibodies from the onset of the disease, the direct link with the severity of the disease and the proportional decrease after treatment [8,31,39]. Anti-type II collagen antibodies are not specific to RP; they have been highlighted in other rheumatic autoimmune diseases, such as rheumatoid arthritis, but have a different epitope specificity [40,41].

The results of murine studies highlight the increased secretion of anti-type II, IX and XI collagen autoantibodies and the development of the clinical manifestations of the disease after immunization with type II collagen. In addition, the important role of genetic predisposition in these cases is supported [19,42]. Taneja’s study included transgenic mice expressing DQ8 in a NOD background, which, after immunization with type II chicken collagen, developed polyarticular damage and auricular inflammation. The humoral response consisted of increased levels of anti-type II, IX and XI collagen antibodies, and the cellular response included T lymphocytes [42]. Similar results were obtained in an analysis of HLA-DQ6/DQ8 double-transgenic mice after immunization with bovine type II collagen [19].

Matrilin-1 is part of the non-collagen proteins of the hyaline cartilage, and it is abundant in skeletal growth cartilages. In adults, it is found only in the nasal septum, ear, trachea and xiphisternal cartilages [43,44]. The important role of matrilin-1 in the pathogenesis of RP was demonstrated in murine studies. Hansson et al. highlighted the development of inflammatory lesions in tracheal cartilage after the administration of anti-matrilin-1 antibodies in adult B-cell-deficient mice [45]. The autoantigen role of this protein in the initiation and progression of RP is supported by an analysis performed on mice expressing HLA-DQ6α8β transgenes, which developed spontaneous polychondritis [46]. A strong IgG humoral response was highlighted, without any anti-collagen type II antibodies [46].

Human studies support the same important role of matrilin-1 in RP pathogenesis and point to its capacity to mediate the interaction between collagen fibers and proteoglycans [47]. The destruction of the cartilage determines the release of matrilin-1, and its serum concentration is increased during periods of exacerbation of the disease [48]. The presence of IgG and IgM anti-matrilin-1 antibodies was found in 13 out of 97 RP patients, and they were associated with respiratory manifestations in 69% of cases. Moreover, seven patients also presented anti-COMPs (cartilage oligomeric matrix proteins) and anti-type IX and XI collagen antibodies [49]. 

COMP (cartilage oligomeric matrix protein) is a protein that is part of the extracellular matrix of the cartilage, and it is also highlighted in tendons and ligaments [50]. The role of COMP is complex, participating in chondrocyte proliferation, collagen secretion and the fibrillation process [51]. Studies have shown that the secretion of anti-COMP antibodies is closely related to the secretion of anti-matrilin-1 antibodies, which cause cartilage destruction [49,52]. Moreover, it seems that COMP participates in the synthesis of the extracellular matrix and in the process of cartilage repair because increased serum titers were highlighted in remission periods of RP, while active disease was associated with a decrease in COMP concentration [48,53]. 

Apart from the autoantibodies presented above, it seems that RP can also be associated with other less specific antibodies, which can be found in various systemic autoimmune diseases. Thus, a study that included 33 RP cases highlighted that 24% of them presented ANCA (antineutrophil cytoplasmic antibody) positivity (perinuclear or diffuse), and their titer correlated with disease activity [54,55]. Also, in RP patients, ANAs (antinuclear antibodies) were found in varying concentrations. When an increased titer of these antibodies is found, an association between RP and another systemic condition is suggested [56]. The presence of anti-corneal epithelium, anti-cochlear and anti-vestibular antibodies was also described [34,57,58]. In patients with RP and central neurological manifestations, such as limbic encephalitis, anti-glutamate receptor (GluR) ε2 (NR2B) autoantibodies were highlighted [59]. 

Figure 2 shows the main autoantibodies associated with RP and the clinical manifestations correlated with their titers.

### 2.4. Cell-Mediated Immune Responses in RP

In the initial stages of the disease, histopathological studies showed the presence of a cellular infiltrate rich in lymphocytes, neutrophils and macrophages at the level of the perichondrium, with the cartilage remaining intact [60,61]. Also, activated antigen-presenting cells were highlighted, and, at the perichondrium–cartilage junction, deposits of C3 and immunoglobulins were found [62]. The most frequent cells encountered were CD68+ monocytes/macrophages and CD4+ Th lymphocytes [63]. With the evolution of the disease, the cartilage was progressively destroyed, showing increased levels of matrix metalloproteinases (MMPs) and proteases, and the chondrocytes were surrounded by lysosomes [62,63]. At the level of perichondral granulations, increased secretions of MMP-8, MMP-9 and elastase were identified, while high levels of MMP-3 and cathepsins L and K were highlighted both in granulations and chondrocytes [62,63].

Innate immunity cells play an important role in RP pathogenesis, both at the onset and in the subsequent evolution of the disease. In the early stages of chondritis, an infiltrate rich in neutrophils was observed, playing a decisive role in the initiation of chondral inflammation [60,61]. Recently, three new inflammatory markers have been proposed to correlate with RP activity, namely, the C-reactive-protein-to-albumin ratio, neutrophil-to-lymphocyte ratio, and platelet-to-lymphocyte ratio [64]. Similar to neutrophil invasion, numerous eosinophils have been identified at the level of biopsies, playing a role in the occurrence of nasal chondritis, in skin or in conjunctival damage [61,65]. Other data support the involvement of mononuclear cells and secreted cytokines in the occurrence of respiratory manifestations in RP [66]. An increase in the mRNA expression of IL-6 (interleukin-6) and IL-1 cytokines was observed, which was associated with the serum level of MMP-3 in admitted patients [66]. 

Moreover, macrophages and neutrophils secrete an increased level of soluble TREM-1 (triggering receptor expressed on myeloid cells-1) in the acute phases of RP [67]. This type I transmembrane receptor that belongs to the immunoglobulin superfamily has high specificity and sensitivity for RP, with the concentration varying depending on the activity of the disease, thus being considered an activity marker [68,69]. The same data support, apart from the increase in TREM-1, a high secretion of other molecules with an inflammatory role, such as MMP-3, VEGF (vascular endothelial growth factor), IFNγ (interferon γ) and CCL4 (C-C motif ligand 4) [68]. 

Another cytokine that controls macrophage activity, namely, MIF (macrophage migration inhibitory factor), has been shown to have an increased serum concentration in RP cases [70]. Also, the increased recruitment of neutrophils and monocytes at the cartilage level is achieved by pro-inflammatory cytokines, such as IL-8, MIP-1β (macrophage inflammatory protein-1β) and MCP-1 (monocyte chemoattractant protein-1), whose serum secretion is greatly increased [71].

Along with all of these, T lymphocytes are the cells with a well-established role in the pathogenesis of the disease, with data from the literature suggesting that RP can be considered a Th1-mediated condition [71]. A specific T-cell response to peptides belonging to matrilin-1 or type II collagen was highlighted [39]. Thus, increased levels of IL-2, IL-12 and INFγ have been found, which correlate with disease activity [71,72]. However, the Th2 cytokine response characterized by the secretion of IL-4, -5, -6 and IL-10 was not associated with RP activity [71,72]. Hu et al. showed that, although the Th1/Th2 ratio is not significantly increased in RP cases, the pathogenesis of the disease is closely related to the Th1 response [73]. 

In RP patients, a reduced level of regulatory T cells (Tregs) was identified followed by a decrease in IL-10 secretion [73,74]. Also, the level of natural killer T cells (NKTs) was lower in these cases, which favors an abnormal immune response and disease progression [75]. Moreover, at the level of CD4+ T cells, the IFNγ/IL-4 ratio was significantly higher [75].

The main cells involved in the initiation and progression of inflammation in RP, secondary to the increased secretion of some specific pro-inflammatory molecules, are exemplified in Figure 3.

## 3. VEXAS Syndrome

The first description of VEXAS syndrome was published in 2020 by Beck and colleagues following whole-exome sequencing in a cohort of 2560 patients [22]. They highlighted mutations in ubiquitin-activating enzyme 1 (UBA1) in three cases of elderly men who presented systemic inflammatory manifestations and cytopenia [22]. A somatic mutation of UBA1 can occur throughout life, not being restricted to a certain type of tissue [76,77]. UBA-1 is the main E1 enzyme that participates in the protein ubiquitylation process characterized by proteasomal degradation [78]. UBA1 has two isoforms: UBA1a, the long form, located in the nucleus, whose translation is initiated by methionine 1 (exon 2), and UBA1b, the short form, located in the cytoplasm, translated by methionine 41 (exon 3) [22,79]. 

The ubiquitylation process participates directly in the turnover of proteins, allowing their sending to the level of proteasomes [80]. It also participates in non-proteolytic events, such as DNA repair, autophagy, inflammation and intracellular signaling [80,81]. The impairment of this process confers clinical heterogeneity, leading to lymphoproliferative manifestations, the appearance of malignancies and auto-inflammatory diseases, or it favors the appearance of infections [82]. 

UBA1b depletion, the cytoplasmic form, is the main cause of the development of VEXAS syndrome, leading to the appearance of an inflammatory phenotype [22,83]. Moreover, in these cases, the presence of mutant myeloid cells, both monocytes and neutrophils, as well as atypical T and B lymphocytes, was highlighted [22]. In VEXAS syndrome, the innate immune system is activated through several pathways. Thus, activated neutrophils and monocytes determine the increased secretion of some pro-inflammatory cytokines such as tumor necrosis factor (TNF), IL-6, IL-8 or INFγ [84]. In addition, atypical neutrophils retain their phagocytosis function and show an increase in the formation of neutrophil extracellular traps (NETs) [85,86]. Other data support the lack of immature B cells due to the presence of an abnormal differentiation of the B line and a numerical increase in monocytes [87]. However, at the blood level, a decrease in the number of monocytes and dendritic cells was described, with the possible causes being the increase in apoptosis or their presence in inflamed areas [88]. Besides all these cellular changes and the inflammatory status, it seems that disease clinical manifestations are also related to the presence of a certain HLA polymorphism, with the data supporting the role of haplotypes, such as HLA-B51 and HLA-27 [89]. 

The clinical manifestations in VEXAS syndrome are polymorphic and progressive, with the inflammation involving all organs and systems. Usually, the disease occurs in adult men over 40 years old, with the average age at onset being 67 years [90]. Although it represents an X-linked somatic syndrome, there are data that also confirm the presence of the disease among women [84,91,92].

The disease has constitutional manifestations, such as recurrent fever, fatigue, weight loss, myalgias or arthralgias [93,94]. All patients present hematological changes characterized by a numerical decrease in cell lines, such as macrocytic anemia, thrombocytopenia, lymphopenia, monocytopenia or neutropenia [95,96]. Moreover, in a high percentage of cases, a myelodysplastic syndrome and bone marrow vacuolization, including erythroid and myeloid precursors, were highlighted [96]. 

Multiple clinical manifestations can mimic various rheumatic autoimmune diseases associated with a myelodysplastic syndrome such as Sweet’s syndrome, RP, polyarteritis nodosa, Behcet’s disease, rheumatoid arthritis, vasculitis of small vessels or even diseases from the group of spondyloarthritis [22,97]. A VEXAS syndrome associated with RP must be diagnosed when the following conditions are met: a male patient with macrocytic anemia and thrombocytopenia below 200 × 103 [24]. Also, the association of the previously mentioned conditions (RP and VEXAS syndrome) is characterized by the presence of more frequent clinical manifestations, such as fever; eye, lung and skin lesions; and a higher level of C-reactive protein [98]. 

VEXAS syndrome is an autoimmune and autoinflammatory condition with multi-systemic clinical manifestations, and the pathogenic process involves all tissues and organs. Thus, patients can present skin, pulmonary, gastrointestinal, cardiovascular, ocular, musculoskeletal or central and peripheral nervous system manifestations. The main clinical signs highlighted in a large percentage of cases are detailed in Table 3.

## 4. Current Therapeutic Strategies in RP and VEXAS Syndrome

Considering that both RP and VEXAS syndrome are systemic autoimmune diseases characterized by the presence of significant inflammation, treatment must be administered as early as possible, with the main objective being the effective control of the inflammatory symptoms. Thus, there are numerous data that support the beneficial role, in both conditions, of corticosteroid therapy and synthetic, biological or targeted synthetic disease-modifying antirheumatic drugs (DMARDs). Currently, there are no treatment guidelines due to the rarity of the disease and the lack of randomized clinical trials, and the choice of therapy is guided by the severity of the clinical manifestations. 

RP therapy is carried out depending on the severity of the clinical manifestations. In mild forms of the disease that present joint pain and inflammation of the cartilage at the level of the external ear and nose, the use of nonsteroidal anti-inflammatory drugs (NSAIDs) in possible combination with dapsone or colchicine is recommended [110,111,112,113]. In severe forms of RP with ocular, cardiac, laryngotracheal, vasculitic or neurological damage, corticosteroids (CSs) are used in medium–high doses, orally or intravenous pulses [114,115]. In serious, non-responsive cases, or in order to induce and maintain disease remission as quickly as possible, synthetic DMARDs (DMARDs) such as methotrexate (MTX), azathioprine (AZA), cyclosporine A or cyclophosphamide (CYC) are used [111,115,116,117,118,119,120]. In the case of an insufficient or absent response, RP therapy includes the use of anti-cytokinic biological agents such as TNFα inhibitors (Infliximab, Adalimumab or Etanercept) and IL-1 (anakinra) and IL-6 (tocilizumab) blockers [115,121,122,123,124,125]. Also, the use of abatacept, an inhibitor of T-lymphocyte costimulation, has proven to be useful in certain cases, unlike rituximab, a B-cell-depleting agent, which does not seem to be effective in RP [126,127]. Besides all this, there are limited data, based mainly on individual clinical experiences, regarding the use, with favorable results, of intravenous immunoglobulins, plasmapheresis, minocycline and 6-mercaptopurine [115,128,129,130].

The therapy of VEXAS syndrome, similar to that of RP, is not yet standardized, with the available results being from clinical case series. Due to the pathogenetic complexity, the disease has a high mortality because of the poor response to many therapeutic agents [131]. A recently published review proposes a treatment algorithm initially based on preventive interventions, with the therapy then being guided by hematological or systemic inflammatory clinical manifestations [132]. Preventive, supportive interventions include anticoagulant treatment and erythrocyte or platelet transfusions, as well as infectious prophylaxis carried out through vaccination and antibiotic or antiviral therapy [132]. Significant systemic inflammation requires the administration of high doses of corticosteroids [131,132]. The management of hematological manifestations involves the use of DNA methyltransferase inhibitors (azacytidine and decitabine), erythropoiesis-stimulating agents, a thrombopoietin receptor agonist—eltrombopag—or targeted synthetic molecules—Janus kinase inhibitors (JAKis—especially ruxolitinib) [102,132,133,134,135,136]. In severe, non-responsive cases, bone marrow transplantation can be used; promising results about allogeneic hematopoietic stem cell transplantation (ASCT) have been published [132,137,138,139]. Systemic inflammatory manifestations, including rheumatological damage, require the use of synthetic immunosuppressants such as methotrexate, azathioprine or cyclosporine, biological DMARDs such as IL-1 (anakinra, canakinumab) and IL-6 (tocilizumab, siltuximab) inhibitors, T-lymphocyte costimulation inhibitors (abatacept) or targeted synthetic DMARDs—JAKis (baricitinib, ruxolitinib) [99,132,140,141,142,143].

## 5. Conclusions

Cartilage is the targeted organ in many autoimmune disorders like RP. Considering all the previously presented data, we can say that RP has a complex pathogenesis that is still incompletely elucidated. Under the influence of external trigger factors, a genetically predisposed person develops the disease. These external factors determine the degradation of cartilage proteins, followed by the release of cryptic cartilage antigens. The latter trigger autoimmunity by recruiting different immune cells and by secreting cytokines with an inflammatory effect, ultimately leading to the destruction of the cartilage matrix and its replacement with fibrotic tissue. Along with RP, VEXAS syndrome, another systemic autoimmune disease with genetic determinism, has an etiopathogenesis that is still incompletely known, and it involves the activation of the innate immune system through different pathways and the appearance of the cytokine storm. The clinical manifestations of VEXAS syndrome include an inflammatory phenotype often similar to that of RP, which raises diagnostic problems. With great care, idiopathic RP must be differentiated from RP associated with VEXAS, based both on the particular clinical manifestations (skin, eye or lung involvement) and on the results of biological tests. Since both diseases have similar pathogenetic mechanisms, there are many common therapies whose main role is to block inflammation and visceral damage. Thus, promising results have been obtained after the use of corticosteroids and synthetic, biological or targeted synthetic disease-modifying drugs. 

## Figures and Tables

**Figure 1 ijms-25-02261-f001:**
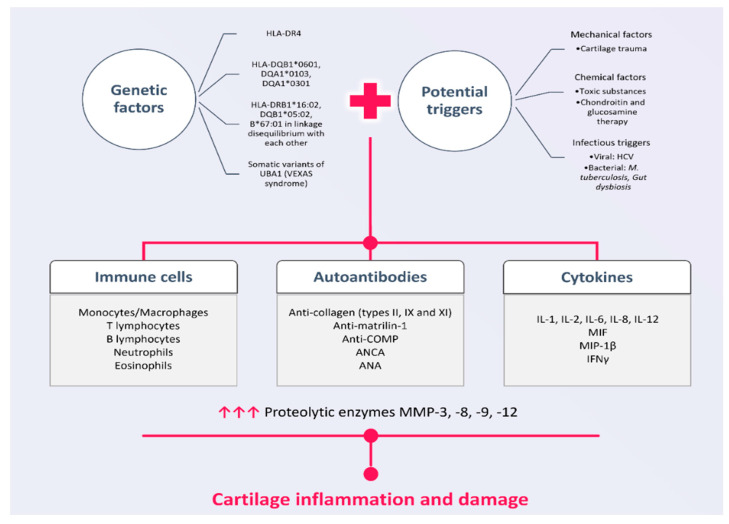
Pathogenic mechanisms involved in the occurrence of inflammation and cartilage damage in RP.

**Figure 2 ijms-25-02261-f002:**
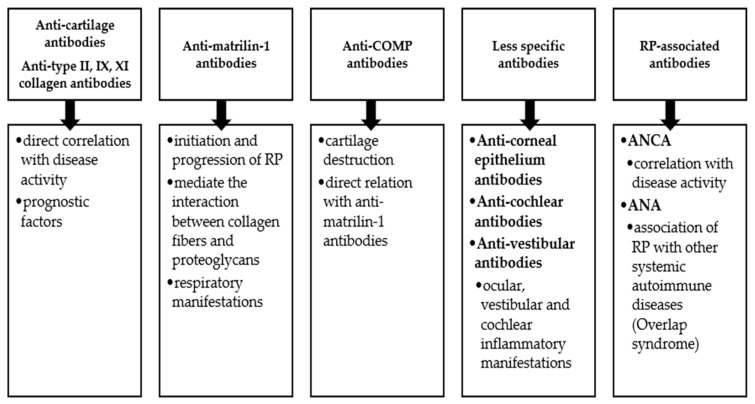
The main autoantibodies in RP. COMP = cartilage oligomeric matrix protein, ANCA = antineutrophil cytoplasmic antibody, ANA = antinuclear antibody, RP = relapsing polychondritis.

**Figure 3 ijms-25-02261-f003:**
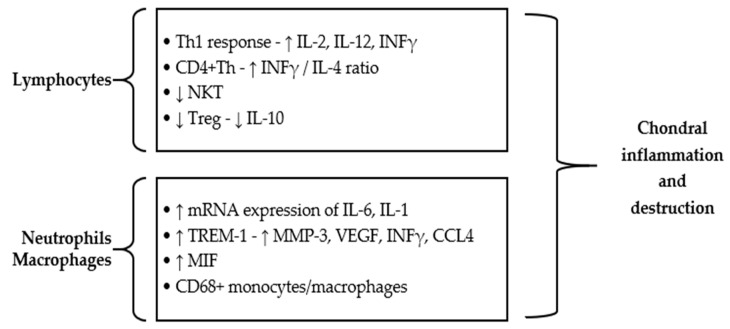
The main cells and molecules with a pro-inflammatory role involved in the initiation of inflammation and cartilage destruction in RP. Th = T helper lymphocyte, IL = interleukin, INFγ = interferon γ, NKT = natural killer T cell, Treg = regulatory T cell, TREM-1 = triggering receptor expressed on myeloid cells-1, MMP = matrix metalloproteinase, VEGF = vascular endothelial growth factor, CCL4 = C-C motif ligand 4, MIF = macrophage migration inhibitory factor, ↑—increased secretion, ↓—decreased secretion.

**Table 1 ijms-25-02261-t001:** Genetic susceptibility in RP.

Genetic Susceptibility in RP (HLA Class II)	The Main Clinical Manifestations	References
HLA-DR4	Major risk of RP occurrence No predominance of a certain subtype	[16,17]
HLA-DRB1 HLA-DQ6/8	Cartilaginous inflammatory manifestations (auricular chondritis)	[16,19]
HLA-DR6	Negative correlations with organ damage A higher median age at disease onset	[17]
DQB1*0601 DQA1*0103 DQA1*0301	Confers susceptibility to RP Severe experimental polychondritis, exhibiting both polyarthritis and auricular chondritis	[20]
HLA-DRB1*16:02 HLA-DQB1*05:02 HLA-B*67:01	Associated with susceptibility to RP (in linkage disequilibrium with each other) Cartilaginous inflammatory manifestations	[21]

RP = relapsing polychondritis, HLA = human leukocyte antigen.

**Table 2 ijms-25-02261-t002:** External triggers involved in the pathogenesis of RP.

Trigger Type		Pathogenic Triggering Mechanism and Clinical Manifestations	References
Mechanical triggers	Trauma to the cartilage	Exposure of cartilage matrix protein antigens	[25]
		Autoimmune response	
	Ear piercing	Inflammatory changes in the nose, ears and upper respiratory tract	[26]
Chemical triggers	Toxic substances (hydrochloric acid, carburetor fluid, waxy internal matrix of a mentholated nasal inhaler)	Nasal and auricular inflammation, peripheral and axial joint damage, scleritis and vestibular disorders	[27]
	After chondroitin and glucosamine therapy initiation	Rapid onset of bilateral auricular chondritis	[28]
	Intravenous papain injection	Ear collapse, tracheal and bronchial damage, even acute respiratory distress	[29]
Infectious triggers	Chronic hepatitis C virus infection Mycobacterium tuberculosis Gut dysbiosis (increase in *Ruminococcus, Bacteroides, Veillonella* and *Eubacterium* species)	Molecular mimicry	[30,31]
		Structural similarity between own HSP and microbial HSP	[32,33]
		Innate immune system activation through the TLR and NLR signaling pathways	[34]
		Cartilaginous inflammatory manifestations	[35]

RP = relapsing polychondritis, HSF = heat shock protein, TLR = Toll-like receptor, NLR = nucleotide-binding oligomerization domain-like receptor.

**Table 3 ijms-25-02261-t003:** Systemic clinical manifestations in VEXAS syndrome.

Organ Involvement	Percentage of Patients	Clinical Manifestations	References
Skin	84%	Dermatitis	[22]
		Cutaneous nodules	[90]
		Vasculitis (medium-vessel arteritis, leukocytoclastic vasculitis)	[95]
		Erytema nodosum	[99]
		Urticaria	[100]
Musculoskeletal	Up to 50%	Arthralgia	[22,24]
		Arthritis	[90,95]
		Myalgia	[99]
		Chondritis (cartilage, ear, nose)	[101,102,103]
Eyes	Up to 40.5%	Episcleritis	[22,23,24]
		Scleritis	[90,95]
		Uveitis	[99]
		Orbital mass	[101,102,104]
		Orbital and periorbital inflammation	[105]
Lungs	Up to 50%	Pulmonary infiltrates	[22,24]
		Pleural effusion	[90,95]
		Pulmonary fibrosis	[99]
		Bronchiolitis obliterans	[101,102,103,104]
Cardiovascular	11%	Pericarditis	[22,23,24]
		Myocarditis	[90,95,99]
		Aortitis	[101,102,103,106]
		Arterial aneurysms	[107,108]
		Venous and arterial thrombosis	
Gastrointestinal	13.8%	Abdominal pain	[22]
		Diarrhea	[90,95]
		Ulcerative lesions	[99,104]
		Digestive obstruction/perforation	[102]
Neurological	Up to 5%	Headache	[22]
		Minor/major cerebrovascular accidents	[90]
		Meningitis	[104]
		Sensory neuropathy	[102]
		Inflammatory demyelinating polyradiculoneuropathy	[109]

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
