# Peer review of "Autoimmunity and Autoinflammation: Relapsing Polychondritis and VEXAS Syndrome Challenge"

_ijms, 2024, doi:10.3390/ijms25042261_

Round 1
Reviewer 1 Report
Comments and Suggestions for Authors
General comments:
The authors provided a novel discussion on autoimmunity and autoinflammation in PR and VEXAS syndrome, chiefly the former.
It is very interesting that these two pathological mechanisms play roles in pathogenesis in two diseases with the disease progression.
Among the spectrum of autoinflammatory, mixed-pattern and autoimmune diseases, why did you focus on only two diseases? You should discuss on not only RP and VEXAS, but also spondyloarthritis, juvenile idiopathic arthritis, Bechet's disease, rheumatoid arthritis, and ANCA-associated vasculitis.
Comments on the Quality of English Language
Line 33: mistyping
Author Response
Dear Reviewer,
Thank you very much for your appreciation and for taking the time to read this article. As you specified and you are completely right, the spectrum of autoimmune and autoinflammatory diseases is very diverse and of particular interest these days. Due to this vastness, we tried to focus only on 2 conditions that seemed interesting to us and, due to their rarity, are less present in the literature. We consider that the title is a general one and may induce some confusion with reference to the content of the work. If you consider it necessary, we undertake to modify the title of the paper, pointing out the 2 conditions that we have presented.
We look forward to your reply.
Thank you,
The authors
Reviewer 2 Report
Comments and Suggestions for Authors The authors have reviewed the pathogenesis and clinical presentation of Polychondritis and VEXAS syndrome. They have compared and contrasted Relapsing Polychondritis (RP) and VEXAS syndrome. This is a review and it summarises existing knowledge, reviews the literature (including work done in humans and animals) and discusses the main pathogenic mechanisms involved. They do not present any original data, nor any of their own cases. It is a useful review because it draws together the published literature on both conditions in a logical well-written format.The authors could include a discussion around management (although they may feel that this is outside the scope of their review).
Figure 1 is helpful as it summarises the pathogenic mechanisms involved in RP. Tabel 1 is helpful as it allows an ‘at a glance’ view of the HLA susceptibility genes in RP and table 2 fulfils a similar role for external triggers involved in the pathogenesis of RP.
They mention the presence of autoantibodies, including ANCA in RP, but don’t specifically discuss how to differentiate RP from ANCA associated vasculitis or other rheumatic diseases. Figure 2 summarizes the autoantibody associations but doesn’t add anything over and above the information already included in the text. Figure 3 is helpful as it summarises a very detailed section on inflammation in a very accessible way.
Less detail is included on VEXAS – although this probably reflects the fact that this was first described in 2020. Tabel 3 summarises the clinical features and is helpful for clinicians. A section on management would be helpful.
The article is comprehensively referenced and clearly written.
Minor Abstract, page 1, line33: prob-lems should be problems
Author Response
Dear Reviewer,
Thank you very much for your appreciation and for taking the time to read this article.
As you very correctly pointed out, we did not discuss the treatment of these 2 conditions. I followed the instructions and added a new chapter that includes data on the management of these 2 conditions.
If you consider it necessary, we can remove figure 2 from the text because it does not provide additional information.
We look forward to your reply.
Thank you,
The authors
Round 2
Reviewer 1 Report
Comments and Suggestions for Authors
The authors should undertake to modify the title, not pointing out only 2 diseases, but all conditions those are considered to be a mixed-pattern with autoinflammation and autoimmunity. The had better state everything to the minutes detail concisely.
Comments on the Quality of English Language
I have no space to tell them.
Author Response
Dear reviewer,
Thank you for the answer. We modified the title according to the instructions received. The new title now specifies in detail the content of the article.
We are at your disposal for any other indications to improve the quality of this work.
Best regards,
The authors